# Effect of Supplemental Inter-Lighting on Paprika Cultivated in an Unheated Greenhouse in Summer Using Various Light-Emitting Diodes

**DOI:** 10.3390/plants12081684

**Published:** 2023-04-17

**Authors:** Yong Beom Kwon, Joo Hwan Lee, Yoo Han Roh, In-Lee Choi, Yongduk Kim, Jidong Kim, Ho-Min Kang

**Affiliations:** 1Interdisciplinary Program in Smart Agriculture, Kangwon National University, Chuncheon 24341, Republic of Korea; nm96727@naver.com (Y.B.K.); ot2581@naver.com (J.H.L.); nuh3722@naver.com (Y.H.R.); 2Agricultural and Life Science Research Institute, Kangwon National University, Chuncheon 24341, Republic of Korea; cil1012@kangwon.ac.kr; 3Cheorwon Plasma Research Institute, Cheorwon 24062, Republic of Korea; ydkim@cpri.re.kr; 4FutureGreen Co., Ltd., Yongin 17095, Republic of Korea; jidong.kim@futuregreen.co.kr

**Keywords:** paprika (*Capsicum annum* L.), summer-cultivated, supplemental lighting, inter-lighting, top-lighting, LED, economic analysis

## Abstract

This study investigated the effects of supplemental inter-lighting on paprika (cv. Nagano RZ) in South Korea in summer using various LED light sources. The following LED inter-lighting treatments were used: QD-IL (blue + wide-red + far-red inter-lighting), CW-IL (cool-white inter-lighting), and B+R-IL (blue + red (1:2) inter-lighting). To investigate the effect of supplemental lighting on each canopy, top-lighting (CW-TL) was also used. Additionally, a control without supplemental lighting was included for comparison. Significant variations were observed in the plant growth indexes 42 days after treatment. The SPAD values and total chlorophyll content in the last period of cultivation were significantly higher than those of the control. In November, the marketable fruit yield was significantly higher than that of the control. QD-IL, CW-IL, and CW-TL resulted in significantly higher values of total soluble solids than the control, and CW-IL resulted in higher values of ascorbic acid content than the control. Regarding the economic analysis, CW-IL resulted in the highest net income rate (12.70%) compared with the control. Therefore, the light sources of CW-IL were assessed as suitable for supplemental lighting due to the highest total soluble solids, ascorbic acid content, and net income rate obtained.

## 1. Introduction

Paprika is a high-income crop and a representative fruit vegetable in greenhouses, and as health-focused eating habits are becoming more widespread, the cultivation area of paprika in Republic of Korea is also increasing [1]. Paprika is cultivated in both summer and winter in Korea. Summer cultivation tends to yield about 20% less than winter cultivation because of intense solar radiation, high air temperatures, and humidity [2]. Intense solar radiation causes high temperatures by increasing the radiant heat in a greenhouse. Exposure to high temperatures in flower buds 16 to 18 days before anthesis causes pollen sterility and a reduction in pollen viability, which reduces fruit size and fruit set [3]. Furthermore, due to summer torrential rain, there is a large variation in solar radiation [4]. In addition, Korea has a summer rainy season called the “Changma” due to the monsoon system in East Asia. Recent weather changes have resulted in heavy rainfall, and increased deep convection was evident in August and September [5,6]. In other words, during the summer cultivation of paprika in Korea, high temperatures caused by intense solar radiation can reduce yields. Additionally, the period of weak sunlight during torrential rain can also be a limitation for paprika cultivation. Previous studies indicated that a decrease in source strength, such as solar radiation, led to a linear increase in the rate of fruit abortion, and it was also reported that a 1% reduction in light resulted in a decrease in the average yield between 0.8 and 1% [7,8]. Shades artificially created to simulate cloudy and rainy days reduced the daily light integral (DLI), which resulted in reduced tomato yields [9]. In addition, fruit vegetables in greenhouses cultivated at high density often lead to excessive mutual shading among plants [10]. Therefore, in order to improve the growth and yield of paprika in greenhouses even in summer, it may be necessary to improve the light environment, not only by shading plants to prevent intense solar radiation but also with supplemental lighting.

Artificial light sources such as a high-pressure sodium (HPS) lamps or light-emitting diodes (LEDs) can be primarily used for supplemental lighting. The choice between HPS lamps and LEDs depends on the application, as HPS lamps have a broader light distribution, which enables a wide area to be covered, and LEDs have a narrower light distribution, with a greater focus on the lighting area [11,12]. However, compared with LEDs, it is difficult to manipulate the spectra of HPS lamps or even to dim them, and they present a great heat emission compared with LEDs [13]. Greenhouse fruit vegetables, such as paprika, mostly use a high-wire training system, i.e., the main stems are supported with a vertical high wire to ensure crop loads [14,15]. Overhead lighting, such as HPS lamps, tends to only focus light on the upper canopy, which can cause mutual shading in the lower canopy and reduce the light reaching the lower parts of high-wire crops [16]. Therefore, in previous studies, supplemental inter-lighting with LEDs has been used in greenhouse fruit vegetables to effectively reduce mutual shading by distributing light to plants [9,16,17,18,19,20,21].

In summer, supplemental lighting can be restricted due to intense solar radiation and high air temperatures. Previous studies reported that supplemental LED inter-lighting during the summertime increased the stomatal conductance and transpiration rate but did not induce physiological changes in the intra-canopy due to the high DLI, and LED inter-lighting during summertime tomato cultivation resulted in a greater dry matter allocation to leaves and stems than to flowering and fruit development compared with the control [22,23]. However, another study reported that although daytime light in the summer could not improve the yield, nighttime LED inter-lighting had a positive effect on photosynthesis, growth and yields in summer and winter [17]. In the Mediterranean and Jordan Valley, which have a relatively high amount of solar radiation, supplemental LED lighting was used for the purpose of increasing yields, while in other studies, it was hypothesized that supplemental LED lighting could improve the yield, total soluble solids, and ascorbic acid parameters of greenhouse tomato plants even in extremely hot summers when shading is needed [19,20,24]. In paprika cultivation in Korea, supplemental lighting has been mostly applied to winter cultivation, which presents low solar radiation [25,26,27]. Nevertheless, the hot period is considered in this approach, and supplemental lighting is used during the rainy season, which has a low amount of solar radiation, so an increase in both quantity and quality can also be expected in summer cultivation.

In horticulture, LED fixtures usually combine red, blue, white, and far-red LEDs; in the case of white LEDs, due to their widespread usage, their fraction can increase to more than 60% of the total LEDs [28]. A well-known method for producing white LEDs is the combination of yellow-emitting yttrium aluminum garnet (YAG) phosphor and blue-emitting LED chips, but this kind of white LED lacks the wavelength of strong red, which is mainly used for photosynthesis [29,30]. Recently, quantum dot (QD) materials were applied in white LEDs to increase the strong-red wavelength, and they were applied in blue LEDs to obtain the blue, red, and far-red wavelengths to provide potential application value for agricultural production [29,30].

In conclusion, it is necessary to determine whether the growth and yield increase through LED supplemental lighting is valid in Korean summer paprika cultivation, which is characterized by high temperatures due to strong solar radiation during the cultivation period and a significant decrease in sunlight due to a long rainy season. Furthermore, even though a 2012 study by Jokinen et al. reported that the overall profitability of LED inter-lighting was highly sensitive to yield advantages, product pricing, and installation costs, it also found that when electricity costs and capital costs were combined, LED product prices were still reported to be too high to become profitable at that time [31]. However, many studies still tend to only compare the effects of LED inter-lighting with physiological effects. Therefore, in this study, we used various light sources to select the most suitable option for Korean summer-cultivated paprika, such as LEDs using QD materials, which provide a customized optimal light quality for production, and different lighting positions such as inter-lighting and top-lighting. Additionally, the profitability of supplemental lighting in summer cultivation was investigated through economic analysis. This study aimed to investigate the effect of supplemental lighting with various LEDs, such as those using QD materials, considering the summer climate in South Korea, which is characterized by high temperatures caused by intense solar radiation and long periods of low light due to the rainy season, and examine the effectiveness of supplemental lighting in summer cultivation using economic analysis.

## 2. Results

### 2.1. Plant Growth

Regarding the plant growth of paprika, except for the leaf area index (LAI), the other values were not significant 56 days after treatment (DAT) (Table 1). However, plant growth index, plant height, number of nodes, number of leaves, and LAI had different initial values for each paprika plant.

The growth indexes were more meaningful in terms of how much they increased during the supplemental lighting treatment compared with the initial value rather than the plant growth state on a specific DAT (Figure 1). Regarding the increases in plant height, number of nodes, and LAI, significant differences were observed between the supplemental-lighting-treated plants and the control at 14, 42, and 28 DAT, respectively, while in the case of the number of leaves, a significant difference was only seen at 42 DAT, and the number of leaves was maintained at the same level in all the experimental groups.

Regarding the number of flowers and number of fruit sets, the number of flowers was maintained at 0~2, and the number of fruit sets was maintained at 5~9 regardless of the supplemental lighting period.

### 2.2. Leaf Characteristics

The SPAD values were not significant at 56 DAT regardless of the canopy, but the NDVI of the mid-canopy showed a significant difference at 56 DAT between the QD and cool-white inter-lighting treatments compared with the control. In the case of Fv/Fm, there was a significant difference among the treatment groups at 56 DAT, but it was difficult to evaluate whether the effect was due to supplemental lighting (Table 2). 

Regarding the variation in SPAD values after treatment, the top-canopy SPAD values were significantly different from those at 0 DAT. Therefore, the results of the leaf characteristics may have varied for each individual leaf. In addition, regarding the results 28 DAT, which were affected by August and September conditions, which did not meet the standard DLI, the average of the SPAD values of the control were lower than those of the supplemental-lighting-treated plants, even though there were no significant differences. However, the difference between the control group and the supplemental-lighting-treated plants at 42 days was not a significant difference. This means that solar radiation had a greater impact than inter-lighting, which compensated for mutual shading due to the plant height (Figure 2a,b). Moreover, at 56 DAT, the paprika was sufficiently grown, and we pinched out the growing tips so that it was able to continuously receive supplemental lighting in a certain position.

The results of the SPAD values and total chlorophyll content measured by harvesting leaves at each height in the last period of cultivation were as follows: Regarding the SPAD values, in the case of the top canopy, the supplemental lighting treatment showed a significant difference compared with the control. In the case of the mid-canopy, the QD inter-lighting treatment showed the highest value, and compared with the control, supplemental lighting resulted in significantly higher values (Figure 2c). The total chlorophyll content, in the case of the top canopy, was significantly higher in the plants receiving all the supplemental lighting treatments, except for those receiving the blue + red inter-lighting treatment, than in control plants; in the case of the mid-canopy, this value was significantly higher in the supplemental-lighting-treated plants, except for plants receiving the cool-white inter-lighting treatment, than in the control (Figure 2d). The significant difference in the measured chlorophyll values seems to have occurred due to the sites for the SPAD value measurement and the sampling sites for the total chlorophyll content measurement. However, regarding both values, the supplemental lighting treatment resulted in higher values than the control. Therefore, to induce a significant difference in the chlorophyll content using inter-lighting, conditions of large mutual shading, such as great plant height or high LAI, and continuous supplemental lighting in the same position to illuminate the same leaves are necessary.

### 2.3. Yield and Fruit Characteristics

In the case of the total fruit set number, there was a difference in September due to the continuous insufficient solar radiation received from August to September. Moreover, in November, the last period of cultivation, the total fruit set number of the supplemental-lighting-treated plants was higher than that of the control plants (Figure 3a).

Marketable fruits refer to fruits excluding those that were dropped or removed due to serious damage and those of less than 100 g in fruit weight. In the case of the marketable fruit number, there were no differences in September, but a significantly higher number was obtained with the supplemental lighting treatment than with the control in November (Figure 3b). Regarding the marketable yield, there was a significant difference between the cool-white top-lighting treatment and the control in November, but no significant differences were found among the other treatments (Figure 3c). The number of non-marketable fruits was the lowest with the cool-white inter-lighting treatment, but this was not significant due to the large standard error between the treatments and blocks (Figure 3d). The marketable yield was the highest with the cool-white inter-lighting treatment, even with the lowest number of fruit sets. Moreover, the lowest number of non-marketable fruits was obtained with the cool-white inter-lighting treatment. Based on these results, there was a difference in the number of fruit sets and the number of marketable fruits obtained with supplemental lighting at harvest in November, at the end of cultivation. Non-marketable fruits can be largely classified into those with blossom rot, insect damage caused by oriental tobacco budworm (*Helicoverpa assaluta*), sunburn, and malformations (Figure 4). Blossom rot occurred in large quantities from August to September, which was a high-temperature period and included a long rainy season. Insect damage occurred in large quantities from September to October. Sunburn and malformations in the paprika fruits rarely occurred. In the case of non-marketable fruit, the degree of occurrence differed for each block and presented large standard errors; thus, the supplemental lighting treatment was not necessarily the cause. However, in the plants receiving the QD inter-lighting treatment and the blue + red inter-lighting treatment with higher blue–light-ratio spectra, there was a tendency of damage due to oriental tobacco budworm. Regarding moths, a higher attractiveness of blue light than white, green, and red light sources was observed [32]. Therefore, it is necessary to check whether the use of LEDs after sunset has any side effects of attracting pests.

There were no significant differences in the physical characteristics of the fruits due to supplemental lighting (Table 3). Among the internal characteristics of the fruits affected by the number of harvest days, the total soluble solids and ascorbic acid content showed significant differences. In the case of total soluble solids, except for the blue + red inter-lighting treatment, the rest of the supplemental lighting treatments resulted in higher values than the control. In the case of the ascorbic acid content, cool-white inter-lighting resulted in the highest value, showing a significant difference compared with blue + red—inter-lighting and the control (Table 4). Due to the high non-marketable rate caused by the blue + red inter-lighting treatment, compared with those observed with other treatments, we did not observe a significant difference compared with the control. In Appolloni’s study (2021), most of the supplemental lighting studies were associated with positive effects on the total soluble solids and ascorbic acid content, but inconsistent results were also found due to other factors [24].

### 2.4. Economic Analysis

The net income rate was the highest with the cool-white inter-lighting treatment due to its high marketable yield and the second-lowest total incremental cost (Table 5). The total incremental costs of LED installation per 1000 m^2^ for the experimental period were calculated to be USD 1769 for QD inter-lighting, USD 2015 for cool-white inter-lighting, USD 2138 for blue + red inter-lighting, and USD 3137 for cool-white top-lighting. For the same period, the electricity usage costs per 1000 m^2^ were calculated to be USD 657 for QD inter-lighting, USD 753 for cool-white inter-lighting, USD 1330 for blue + red inter-lighting, and USD 1137 for cool-white top-lighting. Therefore, almost 70% of the total incremental cost was spent on LED installation.

Since the increase in the gross income due to the increase in the marketable yield obtained with supplemental lighting was confirmed in November, to increase the net income, it is necessary to reduce the total incremental cost using low-cost, high-efficiency LEDs. Additionally, the unit price per kg of green paprika was USD 3.32, which was lower than that of red paprika, whose unit price per kg was USD 3.96. In this experiment, green paprika was sold in November because of concerns about chilling injury. Korean summer cultivation usually involves a cultivation period from June to November. Therefore, it is economically advantageous to distribute the November harvest as red paprika fully ripened in a heated greenhouse.

## 3. Discussion

In this experiment, the internal environment of the greenhouse reached 30 °C in summer, from June to August. In September, the optimum temperature for growth was reached, and in November, the temperature dropped, so cultivation could not be continued because of concerns about chilling injury. In addition, due to torrential rain, the proportions of days when the internal DLI of the greenhouse was less than 12 mol·m^−2^·d^−1^ were 48.4% in August and 60.0% in September. Therefore, in August, the high-temperature period, the occurrence of blossom rot rapidly increased due to the increase in the EC of the medium, and the marketable yield decreased in September, the low-light period.

Inducing significant growth variations in the plant height, number of nodes, and LAI of paprika required supplemental lighting for at least 42 days. The chlorophyll index of the leaf characteristics was measured, and the NDVI showed significant differences for among each of the treatments at 56 DAT. However, SPAD presented both significant and non-significant values during the supplemental lighting period until the pinching out of the growing tips. Therefore, at the end of cultivation, the leaves were harvested to measure their total chlorophyll content together with SPAD. These values were generally higher in the supplemental-lighting-treated plants than in the control plants. In order to increase the chlorophyll content in the leaves with supplemental lighting, at least 56 days were required. In addition, inter-lighting application for a long time under conditions of large mutual shading with an increased plant height or LAI is considered effective for increasing chlorophyll content. In the case of the number of fruit sets, there was a significant difference among the treatments in September, when solar radiation was insufficient, but the difference compared with the control was not large, and there were significant differences in the number of fruit sets, the number of marketable fruits, and the marketable yield compared with November. In terms of the general soluble solids of paprika fruit, the value depends on the period of harvest and the cultivars. Through a comparison of fruit quality among 12 cultivars, it was shown that the range of brix levels was from 6.7 to 9.0 [33]. However, shading has been reported to reduce the soluble solids content of the fruit [9,34]. It has also been reported that higher temperatures during the harvest period result in lower soluble solids contents, and an increase in the soluble solids contents within the fruit is largely due to lower temperatures and assimilated currents [35]. The total soluble solids were generally low due to reduced daily light integral caused by torrential rain and high summer temperatures. However, except for the blue + red inter-lighting treatment, the total soluble solids were significantly higher in the supplemental lighting treatments than in the control. Therefore, summer supplemental lighting can serve as compensation for lower total soluble solids due to an overall harsh environment. However, in the blue + red inter-lighting treatment, the total soluble solids were lower than in the control without supplemental lighting. This may be related to the high levels of non-marketable fruit, such as those with blossom rot, which occurred in the blue + red inter-lighting group around August the most. The general ascorbic acid content of paprika fruit, according to RDA’s Korean Food Composition Database, is known to be 91.75 mg per 100 g, but depending on the cultivars, it can range from 55.3 to 189 mg per 100 g [36]. Supplemental inter-lighting can increase the ascorbic acid content of tomato and paprika fruits [17,37]. However, increasing temperatures above 27 degrees can cause an inhibition of ascorbic acid accumulation [38]. Rather than the effect of shading on the ascorbic acid content, it has been reported that increasing the light intensity can increase the ascorbic acid content and stimulate the antioxidant system [9,39]. In this experiment, the ascorbic acid content in the supplemental-lighting-treated group was higher than that in the control, but only the cool white inter-lighting treatment was significantly different from the control. Therefore, the effect of a high temperature on the ascorbic acid content seems to be greater than that on the total soluble solids. The effect of supplemental lighting on the ascorbic acid content was not as significant as that on the total soluble solids, so the difference is expected to be quite small. In the case of blue + red inter-lighting, there was no difference compared to the control. This was consistent with the tendency of total soluble solids, which is related to the high level and rate of non-marketable fruit, such as those with blossom rot, in August. In addition, the degree of fruit maturity can affect the total soluble solids and firmness, and several reports have shown that LED light can affect the harvest time [40]. Therefore, fruits that were harvested at the same number of days after full bloom were compared. The internal quality, such as the total soluble solids and ascorbic acid contents, showed a significant difference in the supplemental LED lighting treatment. However, there was no significant difference in the firmness, which is related to the cellular texture of the paprika pericarp. The Hunter a value showed no significant difference between the treatments. According to Kim’s research, in the case of paprika fruit color, unlike tomatoes, paprika has an irregular skin surface color; thus, there was no significant difference in color, but the individual carotenoid content was significantly different according to the supplemental lighting [37]. Overall, the effects of supplemental lighting were observed in terms of the growth, yield, and fruit characteristics. However, it was difficult to determine the effects of the supplementing light in a short period of time during the rainy season, from August to September. Inter-lighting is known as an effective supplemental lighting method to eliminate mutual shading and illuminate lower canopies [16]. In this experiment, the observed effects relative to August~September, i.e., the low-light period, are thought to be due to the fact that the plant height or LAI was still too low for mutual shading to occur. In addition, it is thought that the effect of the supplemental lighting was relatively low due to the stress caused by the high temperature, which was above the optimum growth temperature, along with the low-light period. It has been reported that shade-induced stress and high-temperature stress have the same susceptible effect on paprika plants, act with the same process, leading to fruit and flower abscission, and may also act on the assimilates available for flower and fruit development [7]. In this experiment, shade stress was compensated for by using supplemental lighting, but the temperature increase inside the greenhouse could not be suppressed only by shade cloth. Therefore, non-marketable fruits, such as fruits affected by blossom rot, and the rate of fruit dropping were high, even if insect-damaged fruits were excluded. In conclusion, it is considered that it is more effective to perform long-term supplemental lighting in a greenhouse, which is a more controlled environment where year-round cultivation is possible, than using supplemental lighting for short-term cultivation.

Among the LEDs used in this experiment, the QDs had a wide range of red and far-red spectra along with blue spectra. A previous study reported that the effects of far-red supplemental lighting on greenhouse tomatoes in the off season were a high total soluble solids and a high dry matter rate, and the quality was improved enough to be recognized by consumer panelists [41]. In contrast, there was an increase in the yield when supplemental lighting was used in Mediterranean tomato cultivation, but there were no effects of adding far-red lighting at that latitude; in the cultivation of paprika with far-red supplemental lighting, the yield increased but the carotenoid content decreased, indicating an antagonistic relationship [37,42]. On the other hand, there was also a report indicating that the use of far-red supplemental lighting in paprika resulted in dry matter being distributed in the branches and stems and in a reduced number of fruit sets [43]. According to these reports, the effect of far-red supplemental lighting is influenced by various factors, such as the cultivation location and the ratio of far-red light. In this experiment, the treatment with QDs with far-red supplemental lighting resulted in higher values of growth, marketable yield, and total soluble solids than the control, and so did the cool-white treatment. However, the quality of paprika, such as the level of carotenoid pigments in fruits determined according to the wavelength, requires additional confirmation. The cool-white LEDs used in the two treatments of inter-lighting and top-lighting were characterized by a wide green wavelength with a high color temperature. During the experiment, supplemental lighting was applied around sunrise and sunset. At those times, the ratios of the spectrum to the light intensity of the red and green wavelengths were found to be lower than those of the blue wavelengths and those in the daytime. At twilight, the daylight spectrum changes very rapidly and can trigger a strong response in plants [44]. Therefore, it is expected that this effect can be obtained using supplemental lighting with an adequate light spectrum during the sunrise and sunset periods. Green light is known to indicate that the loss of absorptance efficiency due to the sieve effect is small, and it is also known to indicate that the increases in absorptance efficiency are due to the détour effect [45]. In addition, green light drives photosynthesis more effectively than red light in white light at high PPFDs even though red light is greater at low PPFDs [45].

In a research study on paprika treated with supplemental LED inter-lighting in Jordan Valley by Joshi et al. (2019), inter-lighting using cool-white and RGB light resulted in the highest yield, and it was reported that the green spectral component in cool-white LEDs could be advantageous for inter-lighting [19]. In the case of the two cool-white supplemental lighting treatments used in this experiment, the growth, marketable yield, and fruit characteristics were higher than those of the control, but there were no clear differences between the QDs or blue + red lighting and other wavelengths. However, the cool-white inter-lighting induced the highest marketable yield, and the non-marketable fruit rate was relatively low. Therefore, it is necessary to check whether there are side effects, such as an attraction of specific pests (ex. *Helicoverpa assulta*), depending on the spectrum of LEDs when using supplemental LED lighting at sunset.

As a result of the economic analysis, all the supplemental lighting treatments resulted in a higher net income than the control. Among the supplemental lighting treatments, cool-white inter-lighting resulted in the highest net income rate due to the appropriate total incremental costs and the highest marketable yield. It is important to increase the marketable yield to increase the net income rate compared with the control, but it is also important to reduce the total incremental cost through the use of inexpensive and efficient light sources. QD inter-lighting had a lower marketable yield than the two cool-white treatments, but it had the lowest installation costs and electricity costs. Therefore, it is necessary to compare whether it is possible to reduce the costs when manufacturing cool-white wavelength lighting using QD inter-lighting. In particular, top-lighting was very effective in some indexes, such as the November marketable yield and top-canopy leaf SPAD value, but in terms of net income, it resulted in lower values than the QD inter-lighting or cool-white inter-lighting. It seems that more LEDs would be needed for the top-lighting to produce the same intensity as inter-lighting at a certain distance from the paprika plants; as a result, the incremental cost increased, and the net income decreased. Blue + red inter-lighting resulted in the lowest net income among the supplemental lighting treatments. Although non-marketable fruits occurred the most with this treatment, the reason for the low net income seemed to be that the electricity cost out of the total incremental cost was the highest compared with the rest of the light sources.

## 4. Materials and Methods

### 4.1. Plants and Growth Conditions

This study was conducted from 23 May 2022 to 9 November 2022 in an unheated, multi-span plastic greenhouse (width of 13 m, length of 28 m, and eave height of 2.5 m) at Kangwon National University, located in Chuncheon, Gangwon-do (37°52′18.6″ N, 127°44′45.9″ E). The plant material was paprika (*Capsicum annuum* L. cv. Nagano RZ), which is commonly cultivated due to the smallest variation in fruit set between fruit groups, and its high yield and long storage life was used to assess the effect of supplemental lighting on summer cultivation [33,46,47]. The paprika was raised for 4 weeks at Gangwon-do Agricultural Research & Extensions Services and then transferred to a greenhouse. Three paprika seedlings were transplanted at 160 × 33 cm intervals on 100 × 20 × 10 cm Coir slabs (BIOGROW DUO; Biogrow, Mas de la Fabrègue, France). Before transplanting, the Coir slabs were fully hydrated with paprika standard nutrient solution with an EC of 2.0 dS·m^−1^. Irrigation was determined by considering the weather conditions, the growth stage of paprika, the amount of drainage, and the drainage EC. The irrigation EC was between 2.0 and 3.0 ds·m^−1^, and irrigation was performed using 100~150 mL at a time. Irrigation lasted from 1.5~2 h after sunrise up to 3~4 h before sunset. On sunny days, plants were irrigated with irrigation drippers 8~10 times a day. On cloudy days, plants were only irrigated before noon. Paprika was cultivated with twin-head systems that trained the main stem into a “V” shape, branching from 4~5 nodes [15]. In order to maintain the vegetative stage in early cultivation, flowers up to 3 nodes above the branching point were thinned out. To ensure that each node had an adequate number of leaves, one leaf on the main stem and another on each of the lateral stems were kept. When the height of a paprika plant exceeded the greenhouse eave height, we pinched out the growing tips of the shoots.

During cultivation, the internal greenhouse environmental data of air temperature, relative humidity, and integrated solar radiation were monitored in real time with environmental sensors (ioCrops Clima; ioCrops, Seoul, Republic of Korea) placed at the center of the greenhouse. The external greenhouse environmental data were obtained using an automated surface-observing system (ASOS) provided by Korea Meteorological Administration (KMA) (Figure 5a,b). The annual mean value of photosynthetically active radiation (PAR) is 45% of solar radiance, which can be changed by the intensity and photoperiod, and can also be estimated with solar radiance measurements [48,49]. These estimations are particular to the studied region, but the principles can be applied generally [48]. In this study, the estimation of the DLI was based on a report from Korea by Lee et al. (2002) [50] (Figure 5c). The estimated DLI values were compared based on 12 mol·m^−2^·d^−1^, which is known as the minimum DLI required for the paprika production cycle [51]. Regarding the solar spectrum inside the greenhouse, a handheld spectrometer (MK350S; UPRtek, Zhunan, Taiwan) was used to measure the spectrum after sunrise (07:00~08:00), around noon (13:00~14:00), and before sunset (18:00~19:00) at 4-week intervals at the center of the greenhouse, and the average values were used to indicate the wavelength according to the relative intensity (Figure 5d). A 55% aluminum screen was installed in the greenhouse for shading. If the temperature in the greenhouse increased to 30 °C and the solar radiation intensity exceeded 300 W/m^2^, causing the paprika leaves to start wilting, we added the shade cloth. When the solar radiation intensity dropped below 100 W/m^2^, we removed the shade cloth.

### 4.2. LED Fixtures and Supplemental Lighting

Inter-lighting LED light sources were created using 120 cm bar-type LED fixtures with a 40 W power each. We used QDs that included blue- and a wide range of red- and far-red-wavelength LEDs (Cheorwon Plasma Research Institute, Gangwon-do, Republic of Korea), cool-white-wavelength LEDs with a color temperature of 5700 K (HT400-5700; BISSOL LED, Seoul, Republic of Korea), and blue+red-wavelength LEDs with a ratio of blue to red of 1:2 (HT402-1; BISSOL LED, Seoul, Republic of Korea) (Figure 6a). To ensure inter-lighting, we employed a flat hanger bracket to affix two LEDs in opposite directions without an overlap, which illuminated the inner canopy. In addition, they were designed to be height-adjustable with wire so that they could be suspended in the greenhouse. The light intensity was adjusted to around 145 ± 5 µmol·m^−2^·s^−1^ at a distance of 10 cm from the light sources. To determine the effects of supplemental lighting on each canopy, top-lighting LEDs for overhead lighting were produced using cool-white LEDs.

To secure the same level of light intensity as inter-lighting, we arranged three bar-type LEDs without overlapping because the top-lighting structure was at a certain distance from the upper canopy of the paprika. We employed a 300 × 170 mm plastic panel, of which only two ends were bound together to minimize the shading caused by the light sources. In addition, the LEDs were designed to be height-adjustable with wire so that they could be suspended in the greenhouse. Top-lighting LEDs were made to be dimmable to control the light intensity when the height of the LEDs could not be increased because the greenhouse eave height restricted it. The spectrum of each LED fixture used was the average value measured three times during the experimental period. To show the ratio of the wavelengths for each LED fixture, 100% stacked bar graphs of PPF-UV, PPF-B, PPF-G, PPF-R, and PPF-NIR were used (Figure 6b). LEDs for supplemental lighting were installed at 2/3 of the plants based on the slabs between the paprika stems when the paprika plants had grown enough, on 23 July, to illuminate the intra-canopy (Figure 7). Top-lighting was installed 30 cm above the fully developed upper canopy leaves below the growing point, and the light intensity was the same as that of the inter-lighting. Supplemental lighting was applied during 16 h photoperiods, i.e., 04:00~20:00. However, when the greenhouse temperature and solar radiation exceeded 30 °C and 100 W/m^2^, respectively, supplemental lighting was discontinued during the day. Every four weeks, the inter-lighting LED position was adjusted to illuminate the intra-canopy at 2/3 of the plants based on the slabs according to the growth of the paprika plants. Similarly, the top-lighting LED position was also adjusted every four weeks. When the top-lighting LEDs reached the greenhouse eave height and we could not adjust the height of the LEDs, we dimmed the light intensity in alignment with the inter-lighting.

### 4.3. Measurements

The height, number of nodes, LAI, number of flowers, and number of fruits were measured to determine the effect of the supplemental lighting on the paprika growth. The leaf area (LA) of one fully developed leaf each from the top canopy and the mid-canopy was estimated by referring to Lee’s research (2018). The LA of each leaf was calculated as the average of the leaf area of one leaf per paprika plant; then, the LAI was estimated using the formula obtained by modifying Jang’s method (2018) [52,53]:(1)$$\mathrm{LAI}=\bar{\mathrm{LA}}\left(m^{2}\right)\times \frac{\mathrm{No}.\mathrm{of}\;\mathrm{leaves}\left({\mathrm{plant}}^{-1}\right)}{2}\times \mathrm{Stem}\;\mathrm{density}\left(\mathrm{stems}\cdot m^{-2}\right)$$

Measurements were repeated 7 times at 14-day intervals after the supplemental lighting application until the pinching out of the growing tips for each block.

The SPAD value, spectral reflectance parameters, and maximum quantum yield of PS II (Fv/Fm) of the paprika leaves were measured to determine the effect of the supplemental lighting on the paprika leaf characteristics for each canopy leaf. Regarding leaf characteristics, the fully developed leaves below the growing tips were referred to as top-canopy leaves, whereas the leaves illuminated by inter-lighting LEDs were referred to as mid-canopy leaves. The SPAD values for each plant were measured three times, and the average values were used; measurements were conducted on fully developed healthy leaves using a chlorophyll meter (SPAD-502; Minolta Camera Co. Ltd., Tokyo, Japan). The spectral reflectance parameters of the leaves were measured using fully developed healthy leaves with a portable spectrophotometer (Polypen RP 410 UVIS; Photon System Instruments, Drásov, Czech Republic). Among the measured values, the NDVI, which is known to be sensitive to the chlorophyll content, was used [54]. The leaf measurements of the SPAD values and spectral reflectance parameters were repeated 7 times at 14-day intervals after supplemental lighting application until the pinching out of the growing tips for each block. The maximum quantum yield of PS II according to chlorophyll fluorescence was measured 20 min after dark adaptation using a PAM fluorometer (JUNIOR PAM; Heinz Walz GmbH, Effeltrich, Germany). Fv/Fm was measured at the same time in the morning on a sunny day, and 6 repetitions were performed for each block on the 28th, 42nd, and 56th days after the supplemental lighting treatment. At the end of cultivation, the leaves of the top canopy and mid-canopy were harvested for the determination of the SPAD values, and the total chlorophyll content was measured. For measurements, 1 g of finely chopped fresh paprika leaves was dissolved in 10 mL of methanol and extracted at 4 °C for 48 h. Then, the absorbance was measured at 642.5 nm and 660 nm using a UV–vis spectrophotometer (BioMate 3S UV-Vis; Thermo Fisher Scientific, Boston, MA, USA). The measuring method referred to was that reported in Yoon’s research study (2018) [55,56]. The number of measurements for the total chlorophyll contents was 9, 3 measurements for each block.

Paprika fruits more than 60% ripe were harvested at intervals of about 7 days. The harvested fruits were transferred to the laboratory to check their weight and marketability. For the yield per plant and the number of marketable fruits, all the fruits were harvested and calculated separately by month for each block. Marketable fruits were classified as those not affected by blossom rot, insect damage, sunburn, or malformation and those weighing more than 100 g. In the case of November, due to concerns about chilling injury from the drop in temperature in the greenhouse, all fruits, even unripe green peppers, were harvested on November 9th to check the yield per plant, and marketability was also investigated. During cultivation, instances of significant harm resulting from blossom rot and insect damage during fruit development were recorded, and the affected fruits were eliminated to confirm the number of fruit sets and the number of non-marketable fruits. The number of fruit sets and the number of non-marketable fruits were calculated separately by month for each block. The physical characteristics of the fruits such as the yield per plant, length, width, number of locules, and pericarp thickness were investigated. For the physical characteristics, 7 average-sized fruits from a single harvest were selected to obtain an average value representative of that harvest. The average value of each individual harvest was then used as a sample to represent the average value of the entire harvest period. The harvest dates that were not sufficient to be considered representative of a single harvest were excluded from the calculation. Total soluble solids, ascorbic acid content, firmness, and color may change depending on the harvesting period. Therefore, a total of 18 flowers in full bloom, 6 flowers for each block, were tagged three times on 28 July, 15 August, and 30 August, respectively. Then, fruits that were more than 80% ripe were harvested at the same time for comparison. After extracting juice from the fruit, the total soluble solids were measured with a Brix–acidity meter (PAL-BX ACID 1; Atago Co. Ltd., Tokyo, Japan). The number of measurements for the total soluble solids was 21, 7 measurements for each of the 3 harvest seasons. The ascorbic acid content was determined based on the reduction of yellow molybdophosphoric acid to phosphomolybdenum blue. A reflectometer (RQflex plus; Merck, Darmstadt, Germany) and an ascorbic acid tester (Ascorbic Acid Test; Supelco, Bellefonte, PA, USA) were used for the measurements. To obtain the samples for measurement, we put 2 g of fresh fruit pulp in a tube, which we filled up to 20 mL with distilled water, and homogenized it. After centrifugation at 15,000 rpm at 4 °C for 15 min (Mega 17R; Hanil, Seoul, Republic of Korea), the supernatant was obtained and filtered with a 0.45 µm syringe filter (Minisart^®^ Syringe Filters; Sartorius, Göttingen, Germany). Referring to Ribes-Moya’s study (2018), the content of ascorbic acid per 100 g of fruit was measured [57]. The number of measurements for ascorbic acid content was 9, 3 measurements for each of the 3 harvest seasons. For fruit firmness, the paprika pericarp was sliced lengthwise into flat pieces measuring about 30 × 50 mm. The load required to penetrate the pericarp with an 8mm diameter stainless-steel probe was measured using a rheometer (Compac-100II; Sun Scientific Co., Ltd., Tokyo, Japan), and the result was expressed in N (Newton). The number of measurements for firmness was 21, 7 measurements for each of the 3 harvest seasons. Regarding the color, the Hunter a value, which indicates redness, was measured using a color reader (CR-20; Konica Minolta, Tokyo, Japan). The number of measurements for the Hunter a value was 21, 7 measurements for each of the 3 harvest seasons.

Economic analysis. In the economic analysis, we assumed that supplemental lighting was applied for an average of 6.7 h per day for 100 days (3.3 months). Using the yield per plant obtained in this experiment, we assumed a stem density of 6.0 stems·m^−2^ of the cultivation area for 1000 m^2^, and the yield of each supplemental lighting treatment compared with the control without supplemental lighting was investigated. The analysis method was carried out with the calculation reported in Hwang’s research study (2022) [58].
(2)Total incremental cost=LED installation cost+Electricity cost
(3)$$\mathrm{Net}\;\mathrm{income}=\mathrm{Gross}\;\mathrm{income}-\left[\mathrm{Total}\;\mathrm{incremental}\;\mathrm{cost}+\mathrm{Gross}\;\mathrm{income}\;\mathrm{of}\;\mathrm{Control}\right]$$
(4)$$\mathrm{Net}\;\mathrm{income}\;\mathrm{ratio}\;=\left[\;\frac{\mathrm{Gross}\;\mathrm{income}\;\mathrm{of}\;\mathrm{control}+\mathrm{Net}\;\mathrm{income}}{\mathrm{Gross}\;\mathrm{income}\;\mathrm{of}\;\mathrm{control}}\times 100\;\right]-100$$

The LED installation cost share in the total incremental cost was calculated using the unit price at the time of purchase of the LEDs with SMPS, while the electricity cost was determined using a rate of KRW 40 per kWh. The annual cost of paprika per kg used in the calculation of gross income was provided by the Korea Agro-Fisheries & Food Trade Corporation comprehensive agricultural distribution information system (www.nongnet.or.kr). The currency was converted from KRW to USD based on the average exchange rate during the cultivation period (KRW 1342.07 = USD 1).

### 4.4. Experimental Design and Statistical Analysis

The treatments were arranged in a randomized complete block design (RCBD) with three replicates of each treatment to minimize the effect of other environmental factors, and each treatment was applied to 7 plants. The treatments with supplemental inter-lighting were named “QD-IL”, “CW-IL”, and “B+R-IL”, and the treatment with supplemental top-lighting was named “CW-TL”. Additionally, a control treatment (“Cont”) without supplemental lighting was included for comparison. Buffer plants were used between two adjacent treatments to avoid light interactions. Statistical analysis was performed using analysis of variance (ANOVA) with DMRT (Duncan’s multiple range test) at the 5% significance level. Descriptive data were tested in IBM SPSS statistics, version 26.0 (IBM Corp., Chicago, IL, USA).

## 5. Conclusions

This study investigated the effect of supplemental LED inter-lighting on summer-cultivated paprika in Korea. Significant differences were observed among the supplemental lighting treatments in terms of the paprika growth variation, the number of fruits, and the marketable yield, which were generally higher than those of the control plants, which were not treated with supplemental lighting. According to the results of the economic analysis, in the case of the net income rate, compared with the control, the cool-white inter-lighting treatment, which resulted in the lowest number of non-marketable fruits, resulted in the highest net income rate (12.70%), and the rest of the supplemental lighting treatments resulted in net income rates of 3~4%. Thus, summer supplemental lighting is effective. However, for more efficient supplemental lighting, it is expected that it would be more efficient to perform long-term supplemental lighting in a greenhouse where the eave height is high and cultivation is possible throughout the year. In addition, research should continue on lower-cost and more effective light sources, such as cool-white LEDs manufactured with QD materials. This would aim to lower installation and electricity costs while increasing the marketable yield. For future work, it would also be interesting to compare the effect of supplemental lighting treatments among various cultivars of paprika, as each cultivar has different pigments.

## Figures and Tables

**Figure 1 plants-12-01684-f001:**
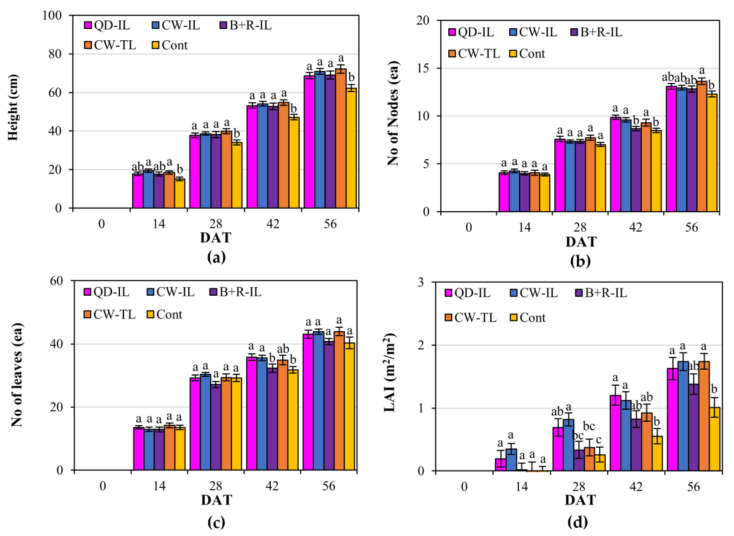
Variance in paprika plant growth during supplemental lighting application. (**a**) Height, (**b**) number of nodes, (**c**) number of leaves, and (**d**) leaf area index (LAI) increased compared with the initial values. Supplemental lighting treatments: QD-IL, QD inter-lighting; CW-IL, cool-white inter-lighting; B+R-IL, blue + red inter-lighting; CW-TL, cool-white top-lighting; Cont, without supplemental lighting. Vertical bars indicate ± SEM (n = 21). Values marked with different letters indicate significant differences according to Duncan’s multiple range test at the 5% level.

**Figure 2 plants-12-01684-f002:**
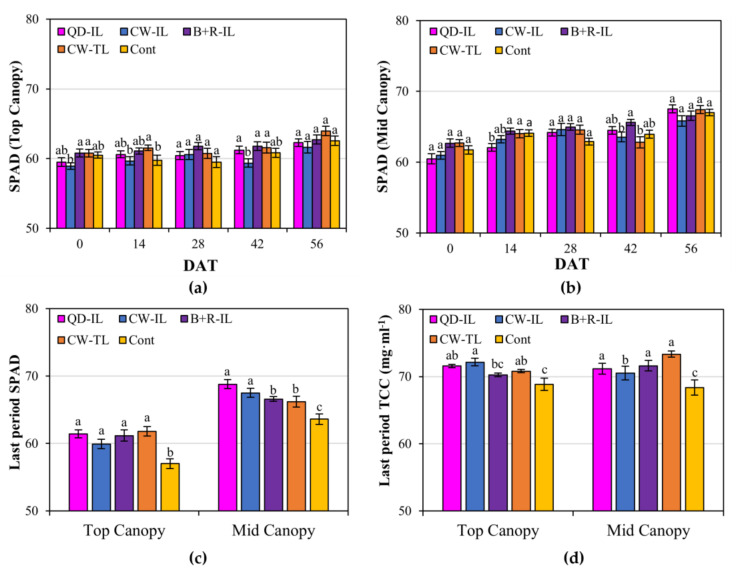
Leaf characteristics, especially chlorophyll content, of paprika plants for each canopy during supplemental lighting application. (**a**,**b**) Variations in SPAD values. (**c**,**d**) SPAD and total chlorophyll content (TCC) of leaves harvested in the last period of cultivation. Supplemental lighting treatments: QD-IL, QD inter-lighting; CW-IL, cool-white inter-lighting; B+R-IL, blue + red inter-lighting; CW-TL, cool-white top-lighting; Cont, without supplemental lighting. Vertical bars indicate ±SEM (n = 21) in (**a**–**c**) and ±SEM (n = 9) in (**d**). Values marked with different letters indicate significant differences according to Duncan’s multiple range test at the 5% level.

**Figure 3 plants-12-01684-f003:**
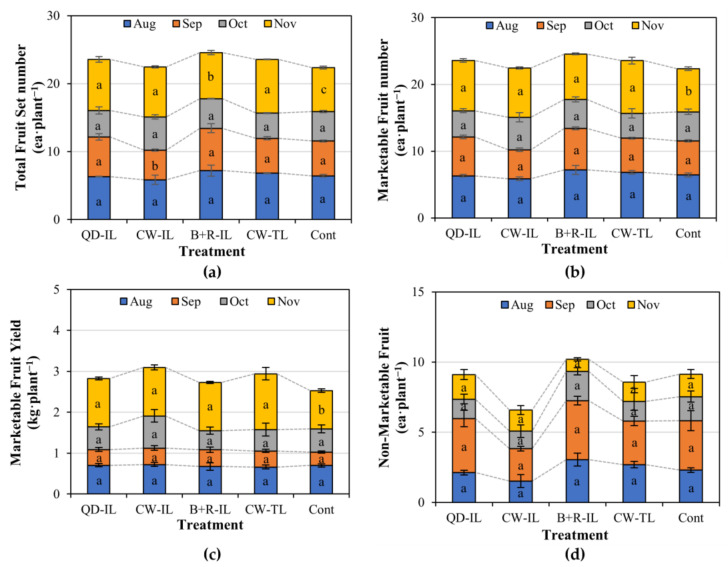
(**a**) Total fruit set number during supplemental lighting application. (**b**) Marketable fruit number during supplemental lighting application. (**c**) Marketable fruit yield during supplemental lighting application. (**d**) Non-marketable fruit number during supplemental lighting application. Marketable fruit means that there were no flaws, such as blossom rot, insect damage, sunburn, or malformation, and the weight was more than 100 g. In November, marketable fruits included green paprika, which was unripe but mature and in equal condition to the marketable variety. Supplemental lighting treatments: QD-IL, QD inter-lighting; CW-IL, cool-white inter-lighting; B+R-IL, blue + red inter-lighting; CW-TL, cool-white top-lighting; Cont, without supplemental lighting. Vertical bars indicate ± SEM (n = 3). Values marked with different letters indicate significant differences according to Duncan’s multiple range test at the 5% level.

**Figure 4 plants-12-01684-f004:**
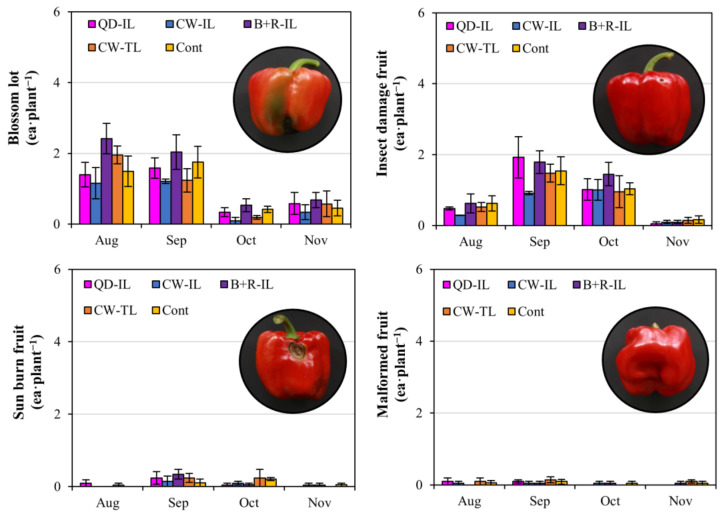
Each type of non-marketable fruit generated during the supplemental lighting period and degree of occurrence. Supplemental lighting treatments: QD-IL, QD inter-lighting; CW-IL, cool-white inter-lighting; B+R-IL, blue + red inter-lighting; CW-TL, cool-white top-lighting; Cont, without supplemental lighting. Vertical bars indicate ±SEM (n = 3). These results were not statistically significant due to the large standard error between each block.

**Figure 5 plants-12-01684-f005:**
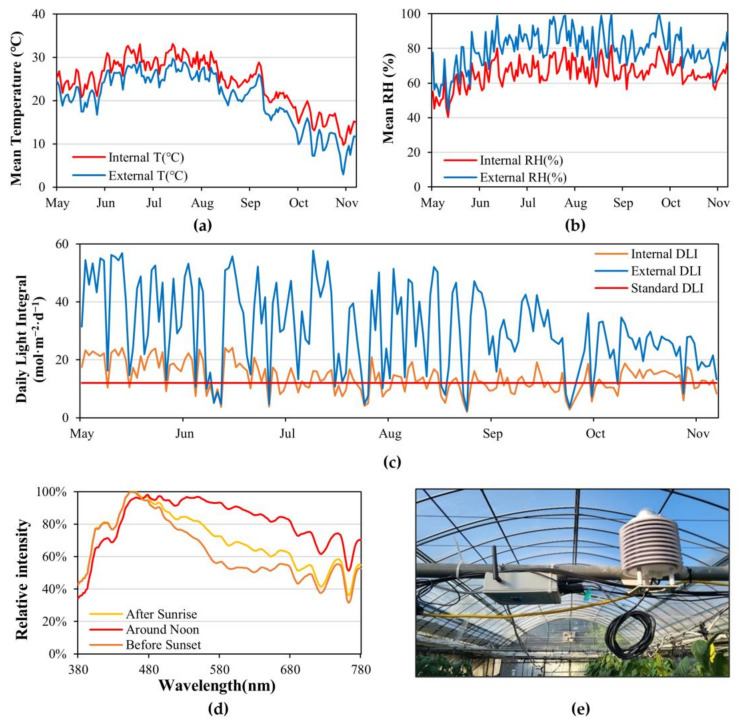
Changes in the greenhouse environment during the experimental period. (**a**) Mean temperature (°C). (**b**) Mean relative humidity (%). (**c**) Daily light integral (mol·m^−2^·d^−1^), where the standard DLI was based on the minimum DLI required for paprika. (**d**) Changes in solar spectrum over time: after sunrise (07:00~08:00), around noon (13:00~14:00), and before sunset (18:00~19:00). (**e**) Image of sensor measuring the greenhouse environment (iocrops Clima).

**Figure 6 plants-12-01684-f006:**
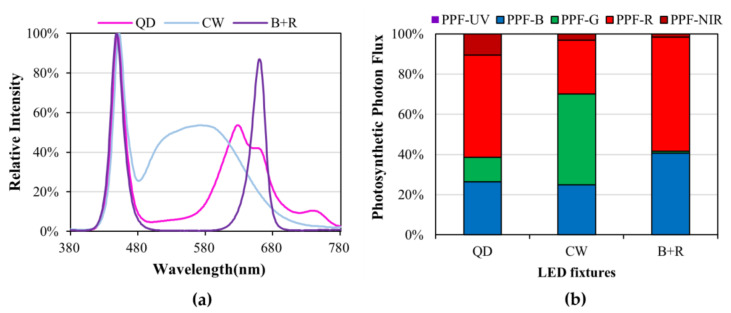
(**a**) Spectrum relative intensity of LED fixtures used in the experiment. QD, quantum dot; CW, cool-white; B+R, blue + red. (**b**) PPF-UV (380~399 nm), PPF-B (400~499 nm), PPF-G (500~599 nm), PPF-R (600~699 nm), and PPF-NIR (700–780 nm) 100% stacked bar graph for the LED fixtures used in the experiment.

**Figure 7 plants-12-01684-f007:**
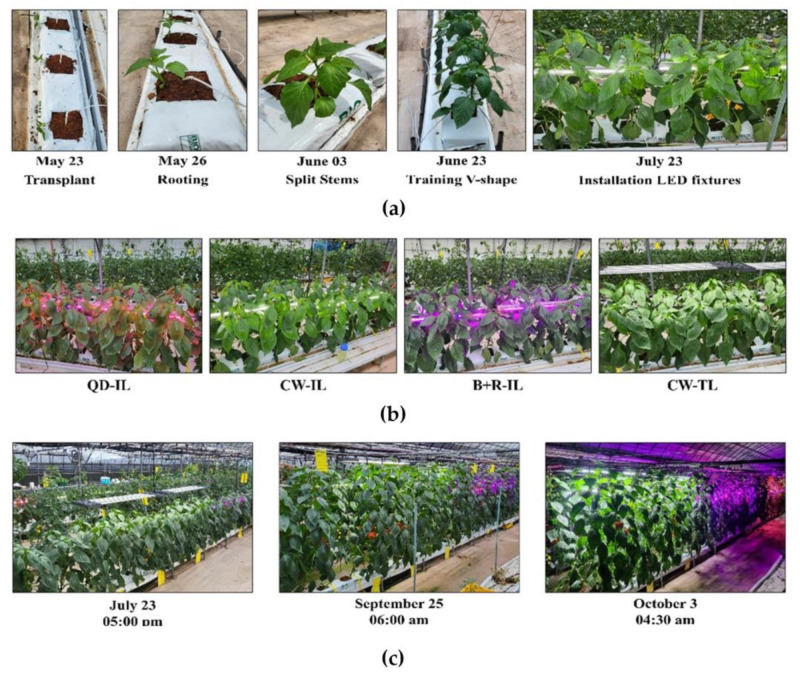
Representative images taken during cultivation. (**a**) Cultivation overview and LED installation. (**b**) Supplemental lighting images for each of the LED treatments: QD-IL, QD inter-lighting; CW-IL, cool-white inter-lighting; B+R-IL, blue + red inter-lighting; CW-TL, cool-white top-lighting. (**c**) View of cultivation sites by period during supplemental lighting application.

**Table 1 plants-12-01684-t001:** Plant growth of paprika after 56 days of supplemental lighting ± SEM (n = 21).

Treatment ^z^	Height (cm)	No. of Nodes (ea)	No. of Leaves (ea)	LAI (m^2^/m^2^)	No. of Flower (ea)	No. of Fruit Set (ea)
QD-IL	155.4 ± 2.0 a ^y^	30.86 ± 0.29 a	91.52 ± 1.32 a	4.15 ± 0.14 ab	2.00 ± 0.12 a	9.49 ± 0.47 a
CW-IL	155.5 ± 1.9 a	30.67 ± 0.23 a	90.62 ± 0.89 a	4.21 ± 0.13 ab	1.71 ± 0.14 a	9.39 ± 0.43 a
B+R-IL	158.0 ± 2.1 a	31.14 ± 0.34 a	90.76 ± 0.88 a	4.18 ± 0.11 ab	1.86 ± 0.11 a	9.71 ± 0.40 a
CW-TL	159.3 ± 2.9 a	31.71 ± 0.34 a	92.48 ± 1.41 a	4.48 ± 0.12 a	2.00 ± 0.13 a	9.38 ± 0.49 a
Cont	151.8 ± 2.3 a	30.52 ± 0.36 a	87.90 ± 1.83 a	3.84 ± 0.14 b	1.67 ± 0.15 a	9.33 ± 0.52 a

^z^ Treatment included: QD-IL, quantum dot LED inter-lighting; CW-IL, cool-white LED inter-lighting; B+R-IL, blue + red LED inter-lighting; CW-TL, cool-white LED top-lighting; Cont, without supplemental lighting. ^y^ Means with different letters within column indicate statistically significant differences by Duncan’s multiple range test at the 5% level.

**Table 2 plants-12-01684-t002:** Leaf characteristics of paprika after 56 days of supplemental lighting ± SEM (n = 21).

Treatment ^z^	SPAD		NDVI		Fv/Fm ^y^	
	Top Canopy	Mid Canopy	Top Canopy	Mid Canopy	Top Canopy	Mid Canopy
QD-IL	62.28 ± 0.56 a ^x^	67.52 ± 0.54 a	0.632 ± 0.003 a	0.637 ± 0.004 a	0.791 ± 0.014 a	0.784 ± 0.007 c
CW-IL	61.60 ± 0.83 a	65.82 ± 0.74 a	0.636 ± 0.003 a	0.633 ± 0.004 a	0.784 ± 0.010 a	0.790 ± 0.008 c
B+R-IL	62.73 ± 0.64 a	66.56 ± 0.67 a	0.629 ± 0.003 a	0.630 ± 0.004 ab	0.774 ± 0.008 a	0.826 ± 0.007 a
CW-TL	63.93 ± 0.67 a	67.41 ± 0.54 a	0.627 ± 0.003 a	0.626 ± 0.004 ab	0.780 ± 0.006 a	0.812 ± 0.005 ab
Cont	62.57 ± 0.68 a	67.01 ± 0.46 a	0.634 ± 0.003 a	0.620 ± 0.004 b	0.774 ± 0.011 a	0.799 ± 0.008 bc

^z^ See Table 1 for details on the treatment included. ^y^ In the case of Fv/Fm (n = 18). ^x^ Means with different letters within column indicate statistically significant differences by Duncan’s multiple range test at the 5% level.

**Table 3 plants-12-01684-t003:** Physical characteristics of fruits during the supplemental lighting ± SEM (n = 30).

Treatment ^z^	Fruit Weight (g)	Fruit Length (mm)	Fruit Width (mm)	No. of Locules (ea)	Pericarp Thickness (mm)
QD-IL	194.4 ± 5.4 a ^y^	90.27 ± 1.55 a	82.52 ± 1.10 a	3.61 ± 0.05 a	6.18 ± 0.11 a
CW-IL	195.4 ± 4.9 a	94.72 ± 1.73 a	82.35 ± 0.94 a	3.49 ± 0.13 a	6.11 ± 0.10 a
B+R-IL	189.0 ± 5.3 a	90.81 ± 1.51 a	80.88 ± 1.07 a	3.61 ± 0.06 a	6.34 ± 0.14 a
CW-TL	192.9 ± 6.0 a	91.22 ± 1.90 a	83.20 ± 1.07 a	3.67 ± 0.08 a	6.05 ± 0.11 a
Cont	197.6 ± 4.8 a	94.58 ± 1.55 a	83.23 ± 1.12 a	3.57 ± 0.07 a	6.21 ± 0.14 a

^z^ See Table 1 for details on the treatment included. ^y^ Means with different letters within column indicate statistically significant differences by Duncan’s multiple range test at the 5% level.

**Table 4 plants-12-01684-t004:** Fruit characteristics during the supplemental lighting affected by harvest time ± SEM (n = 21).

Treatment ^z^	Total Soluble Solids (Brix°)	Ascorbic Acid Contents ^y^ (mg·100 g^−1^)	Firmness (N)	a Value
QD-IL	7.18 ± 0.05 a ^x^	117.9 ± 3.0 ab	37.15 ± 0.67 a	36.28 ± 1.24 a
CW-IL	7.34 ± 0.04 a	121.7 ± 2.3 a	38.52 ± 1.40 a	34.24 ± 0.95 a
B+R-IL	6.87 ± 0.15 b	114.0 ± 1.4 b	35.98 ± 0.97 a	33.37 ± 1.26 a
CW-TL	7.23 ± 0.07 a	117.7 ± 2.8 ab	40.52 ± 1.57 a	34.12 ± 1.02 a
Cont	6.93 ± 0.10 b	111.4 ± 2.2 b	39.41 ± 1.50 a	32.97 ± 0.79 a

^z^ See Table 1 for details on the treatment included. ^y^ In the case of ascorbic acid contents (n = 9). ^x^ Means with different letters within column indicate statistically significant differences by Duncan’s multiple range test at the 5% level.

**Table 5 plants-12-01684-t005:** Economic analysis of paprika fruits during the supplemental lighting.

Treatment ^z^	Marketable Yield (kg/1000 m^2^)	Gross Income ^y^ (USD/1000 m^2^)	Incremental Cost (USD/1000 m^2^)			Net Income ^v^ (USD/1000 m^2^)	Net Income Rate ^u^ (%)
			LED Installation ^x^	Electricity Cost ^w^	Total		
QD-IL	8970	33,262	1769	657	2425	1370	4.65
CW-IL	9660	35,978	2015	753	2768	3743	12.70
B+R-IL	9120	33,857	2138	1330	3468	921	3.13
CW-TL	9480	34,915	3137	1137	4274	1173	3.98
Cont	7890	29,468	-	-	-	-	0.00

^z^ See Table 1 for details on the treatment included. ^y^ Unit price of red or green paprika per kg × marketable yield per 1000 m^2^. ^x^ (LEDs cost + SMPS cost) × number of light sources per 1000 m^2^ ÷ life expectancy of LED ÷ 12 × experiment period. - Life expectancy of LED as 11.42 years, and experiment period as 3.3 months. ^w^ Electricity consumption of light sources × supplemental lighting time × agricultural electricity cost per kWh × number of light sources per 1000 m^2^. - Supplemental lighting time as 6.7 h, and agricultural electricity cost as 40 KRW per kWh. ^v^ Gross income—(total incremental cost + gross income of Control). ^u^ Net income ratio = ((gross income of control + net income)/(gross income of control) × 100) − 100.

## Data Availability

Data is contained within the article.

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
