# Peer review of "Effect of Supplemental Inter-Lighting on Paprika Cultivated in an Unheated Greenhouse in Summer Using Various Light-Emitting Diodes"

_plants, 2023, doi:10.3390/plants12081684_

Round 1

Reviewer 1 Report

The aim of the presented investigations was to study the influence of additional and different LED -exposures on the growth, yield and quality of paprika when grown in a greenhouse. The inclusion of economic aspects, such as income to be achieved, rounds off the picture well. Overall, interesting results have been obtained, but it is a pity that only one variety has been studied. It is known from other studies, e.g. with tomatoes, that the influence of cultivar effects is often more significant than the influence of different exposure systems. It would be good if the authors justified why they conducted their studies on only one cultivar and presented the limitations of the significance of their results. 

The points of criticism in detail:

- Title: abbreviations should not be used in the title; please reword.

- Introduction: the objective should be made clearer. It should be presented which knowledge gaps exist and should be closed with the own experiment. Line 70: which quantitative and qualitative aspects of tomatoes are specifically meant?

- Material and methods: reasons should be given as to why the named cultivar of paprika was selected for the experiment and by what properties it is characterized. Line 369: write Latin name in italics; fruit yield and characteristics: how many fruits were studied in each case? Line 497, 511: fruit hardness is mentioned here, in Table 4 firmness; this is not the same; please choose correct terms and use them throughout. Line 524: area cultivated: the unit of measurement a is not a SI unit, please use SI units only.

- Results: All tables lack information on the number of measurements as well as the standard deviations from the mean values. This also means that it is not always possible to understand why in some cases the treatments do not differ significantly from each other, e.g. in Table 4 for firmness and a-value. Figure 1: "Increase" in the title of all y-axes needs to be removed because it shows the data obtained and only the text explains if it is an increase. Table 4: "Hunter a" as a title for the a color value is unusual, please write a value. 

- Discussion: this chapter should also be further subdivided. Fruit characteristics, such as ascorbic acid and total soluble solids, and firmness, are not discussed. What are common average values and to what extent do the determined data in the control variant as well as then in the exposure variants deviate from them or what are possible causes? Overall, little physiological explanation is given for the effects described.

- Formalia: throughout the text, individual words are repeatedly capitalized in the middle of a sentence, such as paprika; the spelling of subheadings and table headers is also inconsistent and alternates between upper and lower case. This also needs to be standardized.

Author Response

Responses to the editor’s and reviewers’ comment

10 April 2023

Dear reviewers and editorial staff in Plants.

We would like to express our sincere appreciation for your thorough consideration and scrutiny of our manuscript, “Effect of Supplemental Inter-lighting on Paprika Cultivated in an Unheated Greenhouse in Summer Using Various LED Light Sources” Manuscript ID: plants-2325756. Through the accurate comments made by the reviewers, we better understand the critical issues in this paper. We have revised the manuscript according to the reviewer’s suggestions. We hope that our revised manuscript will be considered and accepted for publication in the Plants.

The changes within the revised manuscript were highlighted (in blue). Also, we marked up using the "Track Changes" function in MS Word. After revision, some changes were made to the order of paragraphs to adjust the positioning of figures and tables.

Point-by-point responses to the reviewers’ comments are provided below:

1ST Reviewer

1) Reviewer’s comment:

The aim of the presented investigations was to study the influence of additional and different LED -exposures on the growth, yield and quality of paprika when grown in a greenhouse. The inclusion of economic aspects, such as income to be achieved, rounds off the picture well. Overall, interesting results have been obtained, but it is a pity that only one variety has been studied. It is known from other studies, e.g. with tomatoes, that the influence of cultivar effects is often more significant than the influence of different exposure systems. It would be good if the authors justified why they conducted their studies on only one cultivar and presented the limitations of the significance of their results.

Author’s response:

1) Justify why we use only one cultivar

We appreciate the reviewer's comment. We agree with the reviewer that the influence of cultivar effects is often more significant than the influence of different exposure systems. However, in this study, we compared the effects of supplemental lighting on paprika under special conditions: a Korean summer with too much solar radiation and too little sunlight due to heavy rains. And since profitability was a consideration during the experimental phase, the main point is to use different light sources to select the most suitable one. As you know, for a cultivar to properly express its traits, it must be accompanied by the optimal environment in which it is cultivated. Unfortunately, Korean summer conditions are far from optimal, unlike the cultivation environments in many previous studies where supplemental lighting was used. While there may be some cultivars that perform better in harsh environments, they may not be representative of the typical effects of summer supplemental lighting. There are also some practical difficulties in applying various light sources to different varieties, such as experimental area and costs. Among the references cited is a study that applied a far-red light source to yellow and red paprika to confirm the effects of specific light quality in 2022 [1]. When we apply supplemental lighting to two or more cultivars, it is reasonable to design an experiment to determine the effect of a specific light quality, as referenced above. Therefore, we conducted the present study with one cultivar, "Nagano RZ", which is generally cultivated in Korea. Based on the 1st reviewer's Comment #3 and the 2nd reviewer's Comment #1, we realized that we needed to emphasize the details of the Korean summer climate in the introduction. We also realized that the description of the experimental design, which involved various light sources, needed to be clearer. Therefore, we revised the introduction. Revised Introduction section is shown below:

Line 35: Intense solar radiation causes high temperatures by increasing radiant heat in the greenhouse. Exposure to high temperatures in flower buds 16 to 18 days before anthesis causes pollen sterility and a reduction in pollen viability, which reduces fruit size and fruit set [2].

Line 39: In addition, Korea has a summer rainy season called the "Changma" due to the monsoon system in East Asia. Recent weather changes have resulted in heavy rainfall, and increased deep convection was evident in August and September [3,4]. In other words, during summer cultivation of paprika in Korea, high temperatures caused by intense solar radiation can reduce yields. Additionally, the period of weak sunlight during torrential rain can also be a limitation for paprika cultivation.

Line 95: In conclusion, it is necessary to determine whether the growth and yield increase through LED supplemental lighting is valid in Korean summer paprika cultivation, which is characterized by high temperatures due to strong solar radiation during the cultivation period and a significant decrease in sunlight due to a long rainy season. Furthermore, even though a 2012 study by Jokinen et al. reported that the overall profitability of LED inter-lighting is highly sensitive to yield advantage, product pricing, and installation costs, it also found that when electricity costs and capital costs are combined, LED product prices were still reported to be too high to become profitable at that time [5]. However, many studies still tend to only compare the effects of LED inter-lighting with physiological effects. Therefore, in this study, we used various light sources to select the most suitable option for Korean summer-cultivated paprika, such as LEDs using QD materials, which provide customized optimal light quality for production, and different lighting positions like inter-lighting and top-lighting. Additionally, the profitability of supplemental lighting in summer cultivation through economic analysis.

2) Limitations of our results

We appreciate the reviewer's comment. The supplemental lighting showed a significant difference in growth and yield compared to the control group, which was without supplemental lighting. However, the limitation of the present study is that supplemental lighting is expected to be an effective strategy for longer-term cultivation in a more controlled environment, rather than applying it to short-term summer cultivation with the goal of improving profits (Line 359). Also, we added inter-lighting with quantum dot (QD) LEDs to optimize light efficiency, but the conventional cool white inter-lighting was more profitable. Due to the characteristics of LEDs, which allow for cost reduction through massive production, installation costs need to be confirmed in more detail. If QD LEDs can be used to manufacture LEDs with cool white wavelengths at a lower cost than conventional light sources, it is believed that it will be possible to provide a better and cheaper way of supplemental lighting. If such verification is completed, we believe that it would be meaningful as future work to compare the effects in more detail, such as gene expression of cool white inter-lighting based on QD material that maximizes profitability on various cultivars, as suggested by the reviewer. Therefore, we added a description of the limitations and possibilities of QD LEDs in the discussion, and a description of future work in the conclusion. Revised discussion and conclusion section are shown below:

Line 408: QD inter-lighting has a lower marketable yield than the two cool-white treatments, but it had the lowest installation cost and electricity cost. Therefore, it is necessary to compare whether it is possible to reduce the cost when manufacturing cool-white wavelength lighting using QD inter-lighting.

Line 632: In addition, research should continue on lower-cost and more effective light sources, such as cool white LEDs manufactured with QD materials. This aims to lower installation and electricity costs while increasing marketable yield. For future work, it is also interesting to compare the effect of supplemental lighting treatments among various cultivars of paprika, as each cultivar has different pigments.

2) Reviewer’s comment:

The points of criticism in detail:

- Title: abbreviations should not be used in the title; please reword.

Author’s response:

We appreciate the reviewer’s comment. We have reflected on this comment. Revised title is shown below:

Effect of Supplemental Inter-lighting on Paprika Cultivated in an Unheated Greenhouse in Summer Using Various Light-Emitting Diodes

3) Reviewer’s comment:

- Introduction: the objective should be made clearer. It should be presented which knowledge gaps exist and should be closed with the own experiment.

Author’s response:

We appreciate the reviewer’s comment. In the present study, the knowledge gaps are as follows:

1) It is supplemental lighting applied to the Korean summer environment, which is unsuitable for cultivation due to high temperatures, intense solar radiation, and torrential rain.

2) Compare various light sources, such as QD LEDs with customized wavelengths for plant growth, or LEDs with different light positions, such as inter-lighting and top-lighting.

3) The experiment compared not only growth or yield, but also profitability, which is important from the point of view of practical application.

As discussed in Comment #1, we've revised the introduction section to make it clearer. Please see Comment #1 about the revised introduction section.

4) Reviewer’s comment:

Line 70: which quantitative and qualitative aspects of tomatoes are specifically meant?

Author’s response:

We appreciate the reviewer’s comment. We have reflected on this comment. Revised Introduction section is shown below:

Line 79: ~ while in other studies, it was hypothesized that supplemental LED lighting could improve the yield, total soluble solids, and ascorbic acid parameters of greenhouse tomato plants, even in extremely hot summers when shading is needed [6].

This reference is a meta-analysis type of article, and the parameters presented were those that have been shown to change as a result of supplemental lighting in a common set of studies on tomatoes, a typical fruit vegetable like paprika grown in a greenhouse. Therefore, if there is an effective difference in that parameter, we can assume that the supplemental lighting is effective. It was also hypothesized that LED shading could improve yield, total soluble solids, and ascorbic acid, even in extremely hot summers when shading is required at the conclusion. The Korean summer environment experiences extremely high temperatures, as stated in the reference. Therefore, the present study can serve as validation of the hypothesis presented in that reference.

5) Reviewer’s comment:

- Material and methods: reasons should be given as to why the named cultivar of paprika was selected for the experiment and by what properties it is characterized.

Author’s response:

We appreciate the reviewer’s comment. Based on previous studies, 'Nagano RZ', a well-known red paprika cultivar, is widely cultivated in Korea. Since the experiment is to verify the overall effect of supplemental lighting on summer cultivation.

The characteristics of the Nagano RZ reported in previous studies include the following:

1) In Jang et al.'s study, the red cultivars 'Maranello' and 'Nagano' were assessed to be the two best red cultivars, with the smallest degree of late-season growth inhibition and the smallest variation in fruit set between fruit groups. The balance of growth and yield is important for long-term cultivation [7].

2) In Yeo et al.'s study, they compared the yields of nine cultivars grown from July 27 to November 23, 2020, and found that 'Allrounder' had the highest yields of yellow bell peppers, but 'Nagano' was the best among red bell peppers [8].

3) In Choi et al.'s study, they reported that among the red paprika cultivars, 'Nagano' was the best due to its excessive pericarp thickness, high firmness, low decay rate, and ethylene production, as well as its long shelf life [9].

Therefore, we chose 'Nagano RZ', which is commonly cultivated in Korea, as the plant material. We have revised the material and methods to reflect on this comment. Revised material and methods section is shown below:

Line 426: The plant material was paprika (Capsicum annuum L. cv. Nagano RZ), which is commonly cultivated due to the smallest variation in fruit set between fruit groups, as well as its high yield and long storage life was used to assess the effect of supplemental lighting on summer cultivation [7-9].

6) Reviewer’s comment:

Line 369: write Latin name in italics;

Author’s response:

We appreciate the reviewer’s comment. We have reflected on this comment. The latin name was changed in italics. When revising Comment #5, the corresponding section was also revised.

7) Reviewer’s comment:

fruit yield and characteristics: how many fruits were studied in each case?

Author’s response:

We appreciate the reviewer's comment. In the case of yield, all the fruits are included. However, most of the parameters were averaged to represent a single harvest, and then the averaged values were used to represent the entire harvest. Therefore, it would be more helpful to specify the method and number of measurements rather than the number of fruits used in the study. This is also related to Comment #10. We have more clearly described the measurements in the materials and methods and changed the order of the manuscript to better address the comments. Revised material and methods section is shown below:

Line 552: Harvested fruits were transferred to the laboratory to check their weight and marketability. For yield per plant and the number of marketable fruits, all fruits were harvested and calculated separately by month for each block. Marketable fruits were not affected by blossom rot, insect damage, sunburn, or malformation, and they weighed more than 100 g. In the case of November, due to concerns about chilling injury from the drop in temperature in the greenhouse, all fruits, even unripe green peppers, were harvested on November 9th to check the yield per plant, and marketability was also investigated. During cultivation, instances of significant harm resulting from blossom rot and insect damage during fruit development were recorded, and the affected fruits were eliminated to confirm the number of fruit sets and the number of non-marketable fruits. The number of fruit sets and the number of non-marketable fruits were calculated separately by month for each block. After checking the physical characteristics of the fruits, such as yield per plant, length, width, number of locules, and pericarp thickness were investigated. For physical characteristics, 7 average-sized fruits from a single harvest were selected to obtain an average value representative of that harvest. The average value of each individual harvest was then used as a sample to represent the average value of the entire harvest period. When the harvest dates that were not sufficient to be considered representative of a single harvest were excluded from the calculation. Total soluble solids, ascorbic acid content, firmness, and color may change depending on the harvesting period. Therefore, a total of 18 flowers in full bloom, 6 flowers for each block, were tagged three times on July 28, August 15, and August 30, respectively. Then, fruits that were more than 80% ripe were harvested at the same time for comparison.

Line 576: The number of measurements for total soluble solids was 21 times, 7 times for each of the 3 harvest seasons.

Line 586: The number of measurements for ascorbic acid content was 9 times, 3 times for each of the 3 harvest seasons.

Line 590: The number of measurements for firmness was 21 times, 7 times for each of the 3 harvest seasons.

Line 593: The number of measurements for Hunter a value was 21 times, 7 times for each of the 3 harvest seasons.

We also found that the description of Figure 3 and 4 is incorrect regarding the number of measurements. Not (n=15), (n=3) is correct. Since all fruits were measured, only a single value of yield and the number of fruits per block was obtained. Therefore, a three-replicate experiment was designed to obtain the average. The standard error was obtained from SPSS, so the graph and statistical processing are correct. Only the description text is incorrect.

8) Reviewer’s comment:

Line 497, 511: fruit hardness is mentioned here, in Table 4 firmness; this is not the same; please choose correct terms and use them throughout.

Author’s response:

We appreciate the reviewer’s comment. We've organized the terms into "firmness".

9) Reviewer’s comment:

Line 524: area cultivated: the unit of measurement a is not a SI unit, please use SI units only.

Author’s response:

We appreciate the reviewer’s comment. We reflected on this comment and corrected 10a to 1,000 m2 using the SI unit of m2.

10) Reviewer’s comment:

- Results: All tables lack information on the number of measurements as well as the standard deviations from the mean values. This also means that it is not always possible to understand why in some cases the treatments do not differ significantly from each other, e.g. in Table 4 for firmness and a-value.

Author’s response:

We appreciate the reviewer's comment. We reflected on this comment and added a description of the number of measurements and standard error of the mean to the table. For yield and some fruit characteristics, it is recommended to use the standard error to represent the variability within the entire harvest. Because when we calculate the average value of the entire harvest, we use the average value of a single harvest as a sample of data. We use standard error for consistency in tables and graphs throughout the manuscript.

Also, we revised some of the expressions around measurement to be more clearly defined. Revised material and methods section is shown below:

Line 522: Measurements were repeated 7 times at 14-day intervals after supplemental lighting application until the pinching out of the growing tips for each block.

Line 529: The SPAD values for each plant were measured three times, and the average values were used

Line 538: The leaf measurements of SPAD values and spectral reflectance parameters were repeated 7 times at 14-day intervals after supplemental lighting application until the pinching out of the growing tips for each block.

Line 540: Fv/Fm was measured at the same time in the morning on a sunny day, and 6 repetitions were performed for each block on the 28th, 42nd, and 56th days after supplemental lighting treatment.

Line 549: The number of measurements for total chlorophyll contents was 9 times, 3 times for each of the 3 harvest seasons.

11) Reviewer’s comment:

Figure 1: "Increase" in the title of all y-axes needs to be removed because it shows the data obtained and only the text explains if it is an increase.

Author’s response:

We appreciate the reviewer’s comment. We reflected on this comment and removed "Increase" from the title of all y-axes in Figure 1.

12) Reviewer’s comment:

Table 4: "Hunter a" as a title for the a color value is unusual, please write a value. 

Author’s response:

We appreciate the reviewer's comment. We reflected on this comment and changed the title of Table 4 from “Hunter a” to “a value”.

13) Reviewer’s comment:

- Discussion: this chapter should also be further subdivided. Fruit characteristics, such as ascorbic acid and total soluble solids, and firmness, are not discussed. What are common average values and to what extent do the determined data in the control variant as well as then in the exposure variants deviate from them or what are possible causes? Overall, little physiological explanation is given for the effects described.

Author’s response:

We appreciate the reviewer’s comment. To further subdivide the discussion chapter, the following fruit characteristic parameters have been added with references. Revised discussion section is shown below:

1) Total soluble solids

Line 305: For the general soluble solids of paprika fruit, it depends on the period of harvest and the cultivars. Due to a comparison of fruit quality among 12 cultivars, it was shown that the range of brix levels was from 6.7 to 9.0 [9]. However, shading has been reported to reduce the soluble solids content of the fruit [10,11]. It has also been reported that higher temperatures during the harvest period result in lower soluble solids contents, and the increase in soluble solids contents within the fruit is largely due to lower temperatures and assimilate currents [12]. The total soluble solids were generally low due to reduced daily light integral caused by torrential rain and high summer temperatures. However, except for the Blue + Red inter-lighting treatment, the total soluble solids were significantly higher in the supplemental lighting treatments than in the control. Therefore, summer supplemental lighting can serve as compensation for lower total soluble solids due to an overall harsh environment. However, in the Blue + Red inter-lighting treatment, total soluble solids were lower than in the control without supplemental lighting. This may be related to the high levels of non-marketable fruit, such as blossom rot, which occurred in Blue + Red inter-lighting around August the most.

2) Ascorbic acid contents

Line 319: For the general ascorbic acid content of paprika fruit, according to RDA's Korean Food Composition Database, known to be 91.75 mg per 100 g, but depending on the cultivars, it can range from 55.3 to 189 mg per 100 g [13]. The supplemental inter-lighting can increase the ascorbic acid content of tomato and paprika fruits [1,14]. However, increasing temperatures above 27 degrees, there is an inhibition of ascorbic acid accumulation[15]. Rather than the effect of shading on ascorbic acid, it has been reported that increasing light intensity increases ascorbic acid content and stimulates the antioxidant system [11,16]. In this experiment, the supplemental lighting was higher than the control, but only the cool white inter-lighting treatment was significantly different from the control. Therefore, the effect of high temperature on ascorbic acid content seems to be greater than on total soluble solids. The effect of supplemental lighting is not as significant as total soluble solids, so the difference is expected to be quite small. In the case of Blue + Red inter-lighting, there was no difference compared to the control. This is consistent with the tendency for total soluble solids, which are related to the high level and rate of non-marketable fruit, such as blossom rot, in August.

3) Firmness

In the case of firmness, the method of measurement was varied in each experiment. So, it is difficult to define a general value in the text. In the present study, the load required to penetrate the pericarp with a ∅8mm probe using a rheometer was expressed in Newton. Previous studies comparing the characteristics of 12 paprika varieties using the same methodology as the present study have shown that firmness ranges from 22.1 to 34.11 N, showing different results for different cultivars [9]. The present study has a higher firmness compared to the previous study, with a range of 35.98 to 40.52 N. In other studies, the diameter of the probe or measurement equipment varied, and the units were inconsistent, such as kg·f, g/cm2, which should be understood as relative rather than general values. As the method differs from other experiments, the firmness measurement method has been clarified for easier comprehension. Additionally, the discussion has been modified as follows:

Line 587: For fruit firmness, slice the paprika pericarp lengthwise into flat pieces measuring about 30 × 50 mm. The load required to penetrate the pericarp with a ∅8mm diameter stainless-steel probe was measured using a rheometer (Compac-100II; Sun Scientific Co., Ltd., Tokyo, Japan) and the result was expressed in N (Newton).

Line 333: Also, the degree of fruit maturity can affect total soluble solids and firmness, and several reports have shown that LED light can affect harvest time [17]. Therefore, fruits that were harvested at the same number of days after full bloom were compared. The internal quality, such as total soluble solids and ascorbic acid contents, showed a significant difference in the supplemental LED lighting treatment. However, there was no significant difference in firmness, which is related to the cellular texture of the paprika pericarp.

4) Hunter a value

Line 339: Hunter a value showed no significant difference between treatments. According to Kim's research, in the case of paprika fruit color, unlike tomatoes, paprika has an irregular skin surface color, thus there was no significant difference in color, but the individual carotenoid content was significantly different according to supplemental lighting [1].

14) Reviewer’s comment:

- Formalia: throughout the text, individual words are repeatedly capitalized in the middle of a sentence, such as paprika; the spelling of subheadings and table headers is also inconsistent and alternates between upper and lower case. This also needs to be standardized.

Author’s response:

We appreciate the reviewer's comment. We reflected on this comment and corrected the manuscript. We have indicated the changes in blue on the manuscript.

Reviewer 2 Report

In this study, author investigated the effects of supplemental inter-lighting on paprika (cv. Nagano RZ) in South Korea in summer using various LED light sources. The following LED inter-lighting treatments were used: QD-IL (blue + wide-red + far-red inter-lighting), CW-IL (cool-white inter- lighting), and B+R-IL (blue + red (1:2) inter-lighting). To investigate the effect of supplemental light- ing on each canopy, top lighting (CW-TL) was also used. Additionally, a control without supple- mental lighting was included for comparison. Significant variations were observed in the plant growth indexes 42 days after treatment. SPAD values and total chlorophyll content in the last period of cultivation were significantly higher than those of the control. In November, the marketable fruit yield was significantly higher than that of the control. QD-IL, CW-IL, and CW-TL resulted in significantly higher values of total soluble solids than the control, and CW-IL resulted in higher values of ascorbic acid content than the control. Regarding the economic analysis, CW-IL resulted in the highest net income rate (12.70%) compared with the control. Therefore, the light sources of CW-IL were assessed as suitable for supplemental lighting due to the highest total soluble solids, ascorbic acid content, and net income rate obtained.

This study has practical application value for the planting of paprika. The experiment design is reasonable and the article is clear. However, the logic of introduction is not smooth, so it is suggested to modify it slightly. Here's an example.

1.  Line 34: Summer cultivation tends to yield about 20% less than winter cultivation because of intense solar radiation, high air temperatures, and humidity”.

Line 34: “it was also reported that a 1% reduction in light resulted in a decrease in the average yield between 0.8 and 1%”.

If I understand correctly, it says that in the summer solar radiation is intense the paprika production is low, the latter part of the article is that summer solar radiation is low and Paprika production is low. The article is not well organized, It should be pointed out the intensity of light will inhibit the growth of paprika.

Author Response

Responses to the editor’s and reviewers’ comment

10 April 2023

Dear reviewers and editorial staff in Plants.

We would like to express our sincere appreciation for your thorough consideration and scrutiny of our manuscript, “Effect of Supplemental Inter-lighting on Paprika Cultivated in an Unheated Greenhouse in Summer Using Various LED Light Sources” Manuscript ID: plants-2325756. Through the accurate comments made by the reviewers, we better understand the critical issues in this paper. We have revised the manuscript according to the reviewer’s suggestions. We hope that our revised manuscript will be considered and accepted for publication in the Plants.

The changes within the revised manuscript were highlighted (in blue). Also, we marked up using the "Track Changes" function in MS Word. After revision, some changes were made to the order of paragraphs to adjust the positioning of figures and tables.

Point-by-point responses to the reviewers’ comments are provided below:

2nd reviewer

1) Reviewer’s comment:

In this study, author investigated the effects of supplemental inter-lighting on paprika (cv. Nagano RZ) in South Korea in summer using various LED light sources. The following LED inter-lighting treatments were used: QD-IL (blue + wide-red + far-red inter-lighting), CW-IL (cool-white inter- lighting), and B+R-IL (blue + red (1:2) inter-lighting). To investigate the effect of supplemental light- ing on each canopy, top lighting (CW-TL) was also used. Additionally, a control without supple- mental lighting was included for comparison. Significant variations were observed in the plant growth indexes 42 days after treatment. SPAD values and total chlorophyll content in the last period of cultivation were significantly higher than those of the control. In November, the marketable fruit yield was significantly higher than that of the control. QD-IL, CW-IL, and CW-TL resulted in significantly higher values of total soluble solids than the control, and CW-IL resulted in higher values of ascorbic acid content than the control. Regarding the economic analysis, CW-IL resulted in the highest net income rate (12.70%) compared with the control. Therefore, the light sources of CW-IL were assessed as suitable for supplemental lighting due to the highest total soluble solids, ascorbic acid content, and net income rate obtained.

This study has practical application value for the planting of paprika. The experiment design is reasonable and the article is clear. However, the logic of introduction is not smooth, so it is suggested to modify it slightly. Here's an example.

  1. Line 34: “Summer cultivation tends to yield about 20% less than winter cultivation because of intense solar radiation, high air temperatures, and humidity”.

Line 34: “it was also reported that a 1% reduction in light resulted in a decrease in the average yield between 0.8 and 1%”.

If I understand correctly, it says that in the summer solar radiation is intense the paprika production is low, the latter part of the article is that summer solar radiation is low and Paprika production is low. The article is not well organized, It should be pointed out the intensity of light will inhibit the growth of paprika.

Author’s response:

We appreciate the reviewer's comment. We reflected on this comment and added references that show how strong light intensity inhibits the yield of paprika. However, the point that the authors wanted to emphasize more in the introduction section is that Korea's summer weather is characterized by intense sunlight, but also by a long rainy season with a lack of sunlight. We have added a reference to the climate of the Korean Peninsula to clarify the meaning of this section. Revised introduction section is shown below:

Line 35: Intense solar radiation causes high temperatures by increasing in radiant heat in the greenhouse. Exposure to high temperatures in flower buds 16 to 18 days before anthesis causes pollen sterility and a reduction in pollen viability, which reduces fruit size and fruit set [1].

Line 39: In addition, Korea has a summer rainy season called the "Changma" due to the monsoon system in East Asia. Recent weather changes have resulted in heavy rainfall, and increased deep convection was evident in August and September [2,3]. In other words, during summer cultivation of paprika in Korea, high temperatures caused by intense solar radiation can reduce yields. Additionally, the period of weak sunlight during torrential rain can also be a limitation for paprika cultivation.

References

  1. Erickson, A.; Markhart, A. Flower developmental stage and organ sensitivity of bell pepper (Capsicum annuum L.) to elevated temperature. Plant, Cell & Environment 2002, 25, 123-130, doi:https://doi.org/10.1046/j.0016-8025.2001.00807.x.
  2. Yihui, D.; Chan, J.C. The East Asian summer monsoon: an overview. Meteorology and Atmospheric Physics 2005, 89, 117-142, doi:https://doi.org/10.1007/s00703-005-0125-z.
  3. Song, H.-J. Long-term variations of cloud top patterns associated with heavy rainfall over the Korean peninsula. Journal of Hydrology: Regional Studies 2023, 46, 101337, doi:https://doi.org/10.1016/j.ejrh.2023.101337.

Round 2

Reviewer 1 Report

The reviewer comments have been fully taken into account and the revision of the manuscript has been done very carefully. The manuscript can now be published.